# DR-NASNet: Automated System to Detect and Classify Diabetic Retinopathy Severity Using Improved Pretrained NASNet Model

**DOI:** 10.3390/diagnostics13162645

**Published:** 2023-08-10

**Authors:** Muhammad Zaheer Sajid, Muhammad Fareed Hamid, Ayman Youssef, Javeria Yasmin, Ganeshkumar Perumal, Imran Qureshi, Syed Muhammad Naqi, Qaisar Abbas

**Affiliations:** 1Department of Computer Software Engineering, Military College of Signals (MCS), National University of Science and Technology, Islamabad 44000, Pakistan; msajid.msse-27mcs@student.nust.edu.pk (M.Z.S.);; 2Department of Electrical Engineering, Military College of Signals (MCS), National University of Science and Technology, Islamabad 44000, Pakistan; 3Department of Computers and Systems, Electronics Research Institute, Cairo 12622, Egypt; aymanmahgoub@eri.sci.eg; 4College of Computer and Information Sciences, Imam Mohammad Ibn Saud Islamic University (IMSIU), Riyadh 11432, Saudi Arabia; gpperumal@imamu.edu.sa (G.P.); iqureshi@imamu.edu.sa (I.Q.); 5Department of Computer Science, Quaid-i-Azam University, Islamabad 44000, Pakistan; smnaqi@qau.edu.pk

**Keywords:** vision loss, diabetic retinopathy, deep learning, convolutional neural network, pretrained learning, NASNet, feature extraction, support vector machine

## Abstract

Diabetes is a widely spread disease that significantly affects people’s lives. The leading cause is uncontrolled levels of blood glucose, which develop eye defects over time, including Diabetic Retinopathy (DR), which results in severe visual loss. The primary factor causing blindness is considered to be DR in diabetic patients. DR treatment tries to control the disease’s severity, as it is irreversible. The primary goal of this effort is to create a reliable method for automatically detecting the severity of DR. This paper proposes a new automated system (DR-NASNet) to detect and classify DR severity using an improved pretrained NASNet Model. To develop the DR-NASNet system, we first utilized a preprocessing technique that takes advantage of Ben Graham and CLAHE to lessen noise, emphasize lesions, and ultimately improve DR classification performance. Taking into account the imbalance between classes in the dataset, data augmentation procedures were conducted to control overfitting. Next, we have integrated dense blocks into the NASNet architecture to improve the effectiveness of classification results for five severity levels of DR. In practice, the DR-NASNet model achieves state-of-the-art results with a smaller model size and lower complexity. To test the performance of the DR-NASNet system, a combination of various datasets is used in this paper. To learn effective features from DR images, we used a pretrained model on the dataset. The last step is to put the image into one of five categories: No DR, Mild, Moderate, Proliferate, or Severe. To carry this out, the classifier layer of a linear SVM with a linear activation function must be added. The DR-NASNet system was tested using six different experiments. The system achieves 96.05% accuracy with the challenging DR dataset. The results and comparisons demonstrate that the DR-NASNet system improves a model’s performance and learning ability. As a result, the DR-NASNet system provides assistance to ophthalmologists by describing an effective system for classifying early-stage levels of DR.

## 1. Introduction

The prevalent global disease of diabetes impacts many people’s lives. Blood glucose levels in people with diabetes mellitus tend to rise over time due to the failure of the pancreas to produce or release enough blood insulin [1]. One of the reasons is DR, which ultimately causes permanent vision loss. According to epidemiology studies conducted in Europe, America, Asia, and Australia, there will be 154.9 million people with DR by 2030, up from 100.8 million in 2010, with 30% of them in danger of going blind [2]. DR accounts for 2.6% of blindness causes worldwide [3]. Despite the alarming statistics, research has revealed that more than 90% of new DR cases might be reduced with the right care, vigilance, and monitoring of the eyes.

In DR, the fluid that seeps into the retina from the blood vessels damages it, which could lead to blindness. Classifying DR is a difficult task due to the presence of many components and noises in retinal fundus images. DR is classified mostly by the affected retinal area and the intensity of pathology visible during an eye exam using a slit lamp. The early detection of this disease is important for preserving the condition of the patient’s eye, as this disease is irreversible. Early detection needs an automated system that is mostly based on deep learning algorithms to achieve suitable efficiency. The use of computer-aided diagnostic (CAD) systems in healthcare has received a lot of attention lately. Using the technique of taking color fundus photos can lower the cost of routine screening. Many machine-learning-based approaches have been proposed for DR classification.

Normal or challenging-to-identify symptoms distinguish the early stages of the illness, known as nonproliferative diabetic retinopathy (NPDR). The blood vessels of the retina are compromised in NPDR. Microaneurysms, which are very small blood vessel bulges, can let fluid seep into the retina. Mild NPDR and severe NPDR are subtypes of NPDR. The tiny blood vessels of the retina may exhibit slight, balloon-like swellings in cases where the condition is mild. Several additional blood vessels are blocked in cases of severe NPDR, which causes many regions to lose their blood flow in the retina. Consequently, these retinal regions communicate with the body to help in the growth of new blood vessels. The more severe kind is called proliferative diabetic retinopathy (PDR). Figure 1 shows five various stages of DR.

Researchers have proposed numerous automated methods for the diagnosis of DR as a result of advancements in computer vision technology. Improvements in computer-aided diagnostic (CAD) systems provide several difficulties, including the need to segment blood vessels, identify abnormalities in a retinal picture, and divide the optic disk. Although machine-learning-based systems have demonstrated robust performance in DR detection, their effectiveness depends heavily on custom characteristics, which are extremely challenging to generalize [4]. The development of computer models with many processing layers that can learn data representations at various levels of abstraction is made possible by deep learning (DL). Deep learning has been increasingly popular recently, showing promise in a variety of applications, most notably in image processing, medical image analysis, data analysis, and bioinformatics. DL algorithms have also had a big positive effect on healthcare, because they have led to improvements in applications for screening, recognition, segmentation, prediction, and classification in many areas, such as the abdomen, heart, pathology, and retina. A thorough assessment of deep learning advancements in the field of Diabetic Retinopathy (DR) analysis, including screening, segmentation, prediction, classification, and validation, is provided here in light of the vast body of recent scientific contributions in this field. Deep learning (DL) techniques offer automated feature extraction and classification from fundus pictures to get beyond these restrictions. The main goal of this work is to aid ophthalmologists by describing an effective system for classifying early-stage DR.

To classify the severity level of DR in fundus images, machine learning (ML) and deep learning (DL) models are extensively applied. Nonetheless, these techniques necessitate suitable preprocessing and feature extraction techniques to enhance the results, particularly when the retinography images come from various sources. Generally, these images are captured by various cameras under varying illumination conditions. To mitigate these effects, we implemented an effective color image enhancement technique. Extensive experiments are performed to assess the performance of the proposed model in multiclass recognition of the severity level of DR eye disease. We validate our model, which outperforms state-of-the-art models in multiclass classification tasks, by analyzing the obtained results using a variety of evaluation metrics. For future work, we are considering diversifying and expanding the number of images within the dataset in order to enhance the feature extraction capabilities. For hyperparameter optimization to yield more competitive results, metaheuristic techniques can be utilized. The patient’s familial medical history, daily diet, and nutrition intake can be incorporated into the dataset to provide valuable disease-related information.

### 1.1. Research Motivations and Challenges

Since the majority of the suggested methods concentrate primarily on machine learning (ML), deep learning (DL), and image processing techniques to extract potential features like lesions, hemorrhages, exudates, and cotton-wool spots, these studies neglected to address the variance in scene illumination and light degradation, which has an impact on performance and may lead to biased prediction results. We utilized a dataset with retinographies from many sources for the proposed strategy. As a result, many kinds of noise and distortions may be seen in the images. We seek to investigate the study field of merging image processing and artificial intelligence to create a fully illumination-proof diagnostic tool for DR to address these problems.

A sufficient and variable amount of data is one of the largest obstacles encountered when training a DL or ML model. In DL models, there are many adjustable parameters. The quantity of data should determine the number of these parameters and the difficulty of the endeavor. In the event of disproportion, it has a direct impact on the classification accuracy of the model. This circumstance also contributes to the issue of overfitting. When the model is overfitted and encounters new data, it fails. When classifying a human disease, it is often difficult to acquire sufficient data. Even if there are sufficient data, the data imbalance between classes is another issue. While the amount of data for one or more classes is sufficient, the amount of data for the other class or classes is inadequate. In Section 3, the approach that was used is detailed. However, it is a complex task to detect and classify the severity of Diabetic Retinopathy (DR), which involves several challenges, some of which include the following:(1)Image quality;(2)Large dataset annotation;(3)Class imbalance;(4)Interobserver variability;(5)Feature extraction;(6)Privacy and data security;(7)Clinical validation.

### 1.2. Research Highlights

The following contributions are proposed in this study:(1)A modified NASNet methodology is proposed. To overcome the problem and improve speed, dense blocks were introduced to the NASNet model for classification of severity level of Diabetic Retinopathy.(2)Integrating dense blocks in the NASNet system enhances DR classification by enabling efficient feature reuse, mitigating vanishing gradients, and promoting multiscale feature fusion. The model learns rich representations, improving classification results for different DR severity levels with fewer parameters and better pattern detection.(3)The suggested methodology is trained and tested on a novel dataset, which is itself a big challenge.(4)The use of several preprocessing methods to enhance the usefulness and visibility of retinal images using histogram equalization and Ben Graham’s processing technique.(5)A balanced dataset was created by performing data augmentation techniques on each grade of DR separately to address the lack of annotated data.(6)Various statistical metrics are used to perform comparisons with state-of-the-art DR categorization systems.(7)A linear SVM classifier in integrated with a linear activation function contributes to the final classification of DR severity levels by creating decision boundaries, providing confidence scores, handling multiclass tasks, and leveraging robustness and interpretability. It complements deep features from models like DR-NASNet, leading to accurate predictions for different stages of DR.

### 1.3. Paper Organization

This research paper is categorized as follows: A literature survey related to DR classification is elaborated in Section 2. A brief description of the different datasets that are used in the projected methodology is given in Section 3. The results of the proposed system are described in Section 4. Section 5 contains the discussions of the research study and the results of the proposed methodology. Finally, Section 6 describes the conclusions and future directions of the research.

## 2. Related Work

Many deep learning techniques have been implemented to classify DR. Some of the most recent methodologies have been discussed in the literature. A unique automated feature learning strategy for DR detection using deep learning techniques is presented in [4]. The results show the high effectiveness of their computer-aided approach in providing practical, affordable, and reliable DR diagnostics without relying on physicians to manually review and analyze the images. The proposed pipeline accurately identified aberrant areas in the input fundus images, allowing for clinical assessment and validation of the automated diagnoses. Researchers looked at how deep convolutional neural networks could be used to automatically classify Diabetic Retinopathy using color fundus images [5]. They found that it was more accurate than traditional methods, with a 94.5% accuracy rate on the same dataset used before.

In [6], Hybrid-c, Hybrid-f, and Hybrid-a are the three hybrid types of model structures that enhance the throughput of the DR categorization models through their development together with an enhanced loss function. The fundamental models were EfficientNetB5, EfficientNetB4, Xception, NASNetLarge, and InceptionResNetV2 CNNs. Cross-entropy loss and enhanced cross-entropy loss were used to train these fundamental models, respectively. The hybrid model structures were trained using the output of the fundamental models. The training of the fundamental models could be significantly sped up through the use of enhanced cross-entropy loss, and the models’ performance under different assessment metrics could also be improved.

A novel deep learning methodology is proposed to detect and classify Diabetic Retinopathy [7]. The suggested strategy uses a combination of ResNeXt and a modified DenseNet deep learning model to construct the ensemble model. The CLAHE method is used in the preprocessing of the input photos to balance the histogram. The data augmentation method MixGAN is used to address the class imbalance. The training of the model is performed using the extended dataset. In [8], the Harris Hawks optimization algorithm is used to improve the classification and feature extraction processes for diabetic retinal disease data. Reduced dimensionality and feature extraction are performed with information from the machine learning archive at UCI and a deep learning model that uses principal component analysis (PCA). In [9], convolutional neural networks (CNNs) are proposed to automate the system for DR classification. The system, however, does not achieve high accuracies, and no novel image processing technique was proposed. A CNN tuning using a genetic algorithm is proposed in [10]. A hybrid deep learning model for the DR classification is proposed in [11], where the authors tested the accuracy of different hybrid models along with seven deep learning models. For the hybrid architectures, the authors combined four types of classifiers (SVM, MLP, DT, and KNN) with seven ways to pull out characteristics from deep learning (DL) (DenseNet201, VGG16, VGG19, MobileNet V2, Inception ResNet V2, and ResNet50). The authors used the feature extraction methods used in hybrid designs to complete DL systems. By reducing the memory usage and using a small training set, Macsik et al. [12] emulated the effectiveness of CNN in a way that was suited for systems with constrained memory or processing power. When it came to putting retinal fundus datasets into beneficial and sick groups, they compared an approach with traditional CNN and LBCNN, both of which use probabilistic filter sequences.

Al-Antary and Yasmine [13] proposed a multiscale attention network (MSA-Net) for DR classification. The encoder network improves the resolution of retinal images with mid- and high-level properties and embeds them in a high-level representational space. There are a number of distinctive scale pyramids that describe the retinal structure elsewhere. A multiscale attention methodology enhances feature depiction and discrimination in addition to high-level representation. The model uses cross-entropy loss to classify DR severity. As an extracurricular project, the model uses weak annotations to distinguish between healthy and diseased retinal images. The model is assisted in identifying images of unhealthy retinas by this surrogate task. The proposed method had good results with the EyePACS and APTOS datasets.

In [14], a DenseNet-based architecture is proposed in which images of the fundus are used to train a deep neural network that can do more than one thing at once. The proposed model was trained and tested using the most extensive publicly accessible datasets of retinal fundus images (EyePACS and APTOS datasets). The findings indicate that when compared with the already used five-stage DR classification approaches, the new multitasking model generated the highest performance metrics. The limitations of existing deep learning models are the extensiveness of the datasets employed and the long training times required to train a massive amount of data. A new, innovative deep feature generator patch-based model that demonstrated good classification accuracy for DR is presented in [15]. The proposed method takes cues from the vision transformer (ViT) and multilayer perceptron mixer (MLP-mixer) techniques and separates the picture into vertical and horizontal areas. The unprocessed image and the patches created by deploying pretrained DenseNet201 then generate features. For three-class categorization (normal, PDR, and NPDR), more than 90% accuracy was reached in both the new proposed dataset and the Case 1 dataset made from the APTOS 2019 dataset.

The suggested methodology in [16] used the modified method for removing retinal blood vessels and median filtering on fundus images. A multiclass SVM classifier is implemented to train the model using the retrieved features. Tests are conducted using 1928 fundus images. The proposed methodology has 75.6% sensitivity and 94.5% accuracy on the ‘KAGGLE APTOS’ dataset and 78.5% sensitivity and 93.3% accuracy on the ‘IDRiD’ dataset. Using a single-color fundus image, a unique automatic deep-learning-based method for severity identification is proposed in [17]. The suggested technique employs the encoder from DenseNet169 to create a graphical integration. The Convolutional Block Attention Module (CBAM) is also implemented on the foundation of an encoder to improve the model’s performance. The Kaggle Asia Pacific Tele-Ophthalmology Society (APTOS) dataset is then used to train the model using cross-entropy loss. In total, 98.3% specificity, 97% sensitivity, and 97% accuracy on the binary categorization test were achieved, and a 0.9455 Quadratic Weighted Kappa score (QWK) was achieved, which is superior to other state-of-the-art methods.

A technique for enhancing image quality was implemented in the study [2], which was subsequently merged with a deep-learning-based categorization algorithm. The authors used contrast-limited adaptive histogram equalization (CLAHE) to show that the average accuracy of the method on several models is very good: 91% for the VGG16, 95% for the InceptionV3, and 97% for the EfficientNet. A novel technique for detecting DR using an asymmetric deep learning feature is proposed in [18]. U-Net extracts regular asymmetric deep learning features for segmenting the optic disk and blood vessels. Then, DR lesions are classified using a convolutional neural network (CNN) and a support vector machine (SVM). The lesions are divided into four groups: exudates, microaneurysms, hemorrhages, and normal lesions. Two publicly accessible retinal image datasets, namely APTOS and MESSIDOR, are used to test the proposed methodology. In [19] the development of a revolutionary two-round multilayer severe DR categorization method employing SqueezeNet and a deep convolutional neural network (DCNN) is described. Using SqueezeNet, the fundus picture is first categorized into a normal or anomalous category of DR. In the second-level decomposition, DCNN is used to calculate the severity level of the aberrant images. Table 1 shows various methods used in recent years for DR classification, accuracy, and recall.

Although deep learning models have shown promising results in DR classification, there are still some difficulties that researchers must overcome, such as the need for sizable, annotated datasets, the interpretability of the models, and generalization to multiethnic populations. Overall, the literature indicates that deep learning models can accurately classify DR and can potentially aid in early DR detection and treatment.

## 3. Materials and Methods

Figure 2 depicts the methodology of the proposed model. Initially, the image is acquired using suitable image processing, and then the preprocessing algorithm is used to improve the image visualization. Then, the proposed deep learning model is implemented to detect the features from the images and select the best features appropriate for the DR classification. Finally, the linear SVM (LSVM) classifier is used to categorize the image into distinctive categories. For the categorization of eye-related illnesses like DR, the DR-NASNet system is proposed in this paper. In this system, the NASNet architecture and dense blocks are used. Deep learning is used to extract the useful characteristics of the CAD system. Transformation learning is employed here to retrain a previously learned model on a new dataset, PAK-DR. To extract relevant DR characteristics from retinal fundus images, the CAD-DR architecture has seven major phases, as shown in Figure 2. The properties obtained from both the NASNet design and dense blocks are combined through constituent development. Throughout the training, the thick blocks’ proportions are constantly changing. The classifier layer of SVM with a linear activation function is added as the last step to identify the image as No-DR, Slight, Temperate, Proliferate, and Critical. The addition of the linear activation function smooths the process and enhances the classification outcomes.

### 3.1. Data Acquisition

This procedure entails compiling data from numerous sources, putting it in an appropriate format, and archiving it for future use [20]. Dataset acquisition must follow strict protocols to ensure that the quality of the images is satisfactory for analysis. Factors like image resolution, illumination, contrast, and color balance must be considered to ensure consistency across the dataset. This is the first step in the proposed model, which is very critical for the performance of the CAD-DR system. After acquisition, the images are resized to the appropriate sizes, which are suitable for the deep learning model.

### 3.2. Preprocessing and Data Augmentations

We strengthen the distribution variability of the input pictures by using online data augmentation (DA) during training to strengthen a model’s capability for generalization and resilience. In this work, a variety of widely used augmentation techniques are taken into consideration in order to comprehensively analyze the influence of the composition of data augmentation on DR grading. We use random rotation, random cropping, and horizontal and vertical flipping for geometric modifications. Color distortion, which includes alterations to brightness, contrast, saturation, and hue, is a popular choice for color transformations. One of the essential components for the effective operation of the DL models is the size of the training dataset. To avoid generalization and overfitting problems, a larger dataset must be used to train the deep learning architecture. Although the Kaggle EyePACS dataset size is adequate, when compared with the ImageNet [19] dataset, it is regarded as being quite tiny. Figure 3 illustrates the severely unbalanced distribution of the dataset throughout the classes, with the majority of the photos coming from grade 0. The ratio of this severely unbalanced dataset was 36:37:11, which might lead to inaccurate categorization. We used data augmentation techniques to increase the retinal dataset at different sizes and remove noise from fundus pictures. Due to the dataset’s extreme imbalance, we used distinct data augmentation techniques for each grade in a unique way. A graphic illustration of a few augmentation methods applied to already processed photos is shown. The parameters are mentioned in Table 2.

Preprocessing image data is essential, since the categorization outcomes are influenced by how well the images are preprocessed, as described in Algorithm 1. Images were blurred using the Gaussian blur method in the initial stages of preprocessing to decrease noise. Ben Graham is the name given to the blurring technique, since he recommends the blurring parameter. The image was blurred, and then CLAHE was used to boost the contrast. Prior to CLAHE, blurring was employed to prevent CLAHE from amplifying the undesirable sounds in the photographs. The results of preprocessing are depicted in Figure 4. There are five pictures from five courses. The photographs in the first column are those without any preprocessing, shown in Figure 3, while the images in the second column are those in Figure 3. The categorization of the lesions [19] determines the DR phases, and the last column shows the contrast-enhanced retinograph images. Figure 3 shows the overall result obtained by preprocessing retinograph images, and the corresponding color histogram in Figure 4 shows the intensity distribution of the pixels.
**Algorithm 1.** Contrast and Illumination adjustment by BenGraham and CLAHE techniques.InputOutputRead two-dimenional input fundus image Enhanced contrast and adjust illumination steps are preprocessed retinographics images.Step 1 Img = Gaussian blur (Read (224 × 224)) pixels image Step 2Enhanced-image = Ben-Graham (Img)Step 3(a) blurred_img: Apply Gaussian Blur on img (Gaussian Kernel Size = (0,0), SigmaX = 9) Step 4(b) blend_img: Blending Blurred Image with Original Image (Source1 = img, alpha = 4, source2 = blurred_img, beta = −4, gamma = 128)Step 5Create CLAHE: Clip_Limit = 2, Tile Size = (8, 8) Step 6final_img: Apply CLAHE on blend_img
[End of Algorithm]

In this investigation, histogram equalization (HE) increased model precision and picture contrast. It is mostly used for low-contrast photographs taken for scientific purposes, such as X-ray, thermal, and satellite images [15]. An image contrast enhancement is also important for this inquiry due to the poor contrast of DR pictures. The CLAHE method of HE was employed in this investigation. It is an adaptive histogram equalization (AHE) version. The AHE, however, overamplifies the noise in the area where the picture is essentially homogeneous [16]. The solution, limiting amplification in CLAHE, solves the issue. The amplification is clipped by the clipping factor. In this investigation, a clipping factor of 2 and a tile size of 88 were employed. The fact that the photos are now in grayscale with a single channel after applying Ben Graham to FIs was also taken into account. The single channel was replicated into three channels in order to execute CLAHE on these processed photos, which are displayed in Figure 4. Since CLAHE works with BGR images, the graphic demonstrates how the teachings are clearer after using CLAHE.

### 3.3. Improved NASNet Pretrained Model

The proposed deep learning model consists of three main components. These components are connected together in the sequence shown in Figure 5 and described in Algorithm 2. An Enhanced NASNet model is proposed in this research to enhance the precision of DR classification. Based on the current NASNet concept, this model employs dense blocks to boost efficiency. Although the NASNet, based on convolutional neural network design, may determine the ideal network architecture for medical image analysis, it faces some challenges related to accuracy. To overcome the aforesaid problem and improve speed, dense blocks were introduced to the NASNet model. Moreover, better feature reuse enables the model to extract more relevant data from each input image.

Dense blocks help the deep neural network accurately learn complicated features from the input data. By using these blocks, deep neural networks use fewer parameters overall, which reduces the risk of overfitting and speeds up training time. Dense blocks have become popular in DL due to their ability to extract valuable information from large amounts of complex data. Figure 4 shows the block diagram of dense blocks with five layers and a progression rate of k = 4.

In addition, the training of the model is conducted using a weighted loss function that compensates for class discrepancies in the DR dataset. As a result, its sensitivity and accuracy for detecting DR will increase. The improved NASNet architecture is illustrated below in Figure 6. Several convolutional layers, pooling layers, and activation functions are added to the image to build the NASNet model. The output of each layer is relayed as input to its surrounding layer until the final layer generates a label.
**Algorithm 2.** Proposed DR-NASNet system with preprocessing steps.RequiredFundus Images and Labels (X, Y)Step 1Dataset acquisition, i.e., Fundus Images x ε XStep 2PreprocessingData AugmentationImage Enhancement Step 3Load NASNet Mobile Model ImageNet Pretrained WeightsConvolutional LayersPooling Layers and Activation FunctionsStep 4Introduction of 6 Dense Blocks after Layer 9 of NASNet ModelStep 5 Introduction of Skip Connections Step 6Use of the Flattened Layer, the Feature Mapx = (x1, x2,……… xn)Step 7Model Evaluation 

Let us assume Y matches output picture X’s class label. We obtain a representation of the NASNet model from a function f(X) that transfers the input picture to its corresponding class label:(1)Z=f(x)

Through a training process that minimizes a loss function L(*Z*, *Y*′), which gauges the disparity between the expected output *Z*′ and the actual output *Z*, the parameters of this function are discovered. A logarithmic function is used to generate the loss function L(*Z*, *Z*′), which penalizes inaccurate predictions more severely than correct ones. In more detail, the disparity between the actual label $Z_{i}$ and the anticipated label ${Z_{i}}^{\prime }$ is computed for each class label i in the output vector. This difference is zero if the forecast is accurate ($Z_{i}=1$); otherwise, it is positive and inversely proportional to the prediction error. The total difference between *Z* and *Z*′ can be calculated by adding together these differences over all class labels and taking the negative logarithm of the result. This can be mathematically stated as
(2)$$L\left(z,z^{\prime }\right)=-\sum_{n}^{i=0}Z_{i}\log{z^{\prime }}_{i}$$

The sum of all *i* class labels in the output vector is calculated using this formula, where *Z* is the actual label vector (each element of which indicates whether or not a specific class applies to the input image), and *Z*′ is the predicted label vector (each element of which indicates the likelihood that a specific class applies to the input image). The logarithmic function makes sure that the loss rises exponentially as the predicted probabilities drift further away from the actual labels, promoting the model’s ability to forecast outcomes correctly. Backpropagation is a method used to change the parameters of f(X) throughout the training phase. This entails calculating the gradient of the loss function with respect to each model parameter and changing those parameters in a way that lowers the loss. A learning rate hyperparameter, which controls how much each update impacts the final parameters, regulates the magnitude of these parameter updates.

The NASNet model employed in the study may be used to predict class labels for new images once training is finished. In doing so, the image is fed into the model, which generates a predicted label vector *Z*′. Then, this vector can be used to identify the classes for the input image that are most likely. When utilizing deep learning models like NASNet, it is vital to keep in mind that they need a lot of labeled data to train well. These models can be computationally complex to create and train, frequently necessitating specialized equipment like graphics processing units (GPUs). However, despite these difficulties, deep learning has demonstrated enormous promise in a range of applications and is probably going to keep advancing artificial intelligence for years to come.

### 3.4. Severity Classification by Support Vector Machines

The automatic classification of HR is accomplished using a 75% to 25% train–test separating method and a linear SVM machine learning classifier. The steps are briefly described in Algorithm 3. Due to its efficiency and ability to manage limited datasets, linear SVM is widely utilized. SVM is a classification technique for machine learning that outperforms other classifiers and is frequently used to tackle practical problems. To increase the effectiveness of the proposed method and find the ideal hyperplane that separates the characteristic space of sick and normal cells in the images of retina, we also used linear SVM. Typically, an LSVM inputs a vector *Y* = (*z*1, *z*2,…, *z*n) and outputs a value:(3)$$Y_{out}=\left(Weight,\;Z_{v}^{i}\right)+c$$
where *Weig* stands in for the weight and *c* for the offset; both terms are learned during training. *Ziv* is the input vector, and depending on whether y is greater than or less than 0, it is either assigned to class 1 or, based on whether the variable y is more than or lesser than 0, class 1 will apply. To obtain multiclass output, we utilized a 1-versus-all approach.
**Algorithm 3.** Classification of severity level of DR using retinograph images.
Input Extracted feature map x = (*z*1, *z*2,…., *z*n) with annotations x = 0.1, Test data ZtestoutputClassification of PL disease to normal and abnormalStep1Initially, the classifier and Kernel Regularize L2 parameters are defined for optimization.Step2LSVM ConstructionLSVM training using extracted features *x* = (*z*1, *z*2,…, *z*n) by Algorithm 1.Hyperplane generation $X_{i}=\frac{X_{i}-\mu B}{\sigma^{2}B+\varepsilon }$
*X_i_* batch minimum activating value, *μB* mini batch mean, *σ*^2^*_B_* mini-batch variance, *ϵ* constant added for numerical stabilityStep3The class label is allocated for Testing samples using the decision function of the equation as: Ztest = (Weig, Ziv) + c


## 4. Experimental Results

A different dataset was tested to demonstrate the deployed DR-NASNet DL system’s efficacy and compare results with state-of-the-art techniques. Following the proposed training technique, the dataset was divided into three categories: Eighty percent of the data were used for training (9952 photographs), ten percent were used for testing (1012 photographs), and the remaining ten percent were randomly selected and used as a validation set (1025 photographs) to evaluate performance and store the optimal weight combinations. Overall, the distribution of images is described in Table 3. All photographs were scaled down to a resolution of (224 × 224 × 3) pixels during the training procedure. We evaluated the TensorFlow Keras implementation of the proposed system on a Windows desktop with a graphical processing unit (GPU) and 8GB of RAM.

The proposed framework was initially trained using the Adam optimizer and a method that slows down training when learning has halted for too long (i.e., validation patience) on the retinographic dataset. The following hyperparameters were inputted into the Adam optimizer throughout the complete training process: In this simulation, we used a range of 1E3 to 1E5 for the learning rate, 2–64 for the batch size (with a 2 increase), 50 epochs, 10 for tolerance, and 0.90 for momentum. Our several distinct measures are rounded out by a technique known as “batching” for the distribution of forms.

This study details the evaluation procedures and their outcomes. Classifier precision (Acc) is a common performance metric. It is calculated by dividing the number of correctly classified examples (images) by the total number of examples in the dataset (Equation (1)). Precision (PR) and recall (RE) are commonly used to evaluate image classification systems. As demonstrated by Equation (2), precision increases as the number of correctly labeled images increases, whereas recall is the proportion of correctly categorized images in the dataset to those that are numerically related (3). In addition, the F1-score indicates that the system is more accurate at anticipating the future than systems with lower values. Formula (4) computes the F1-score. The final criterion of this study, top N accuracy, requires the highest probability answers from model N to match the expected softmax distribution. If one of N’s predictions corresponds to the desired label, then the classification is accurate.
(4)$$Accuracy\left(ACC\right)=\frac{TP+TN}{TP+TN+FP+FN}$$
(5)$$Precision\left(PR\right)=\frac{TP}{TP+FP}$$
(6)$$Recall\left(RC\right)=\frac{TP}{TP+FN}$$
(7)$$F1-score=2\times \frac{PR\times RC}{PR+RC}$$
(8)$$MCC=\frac{TP\times TN-(FP\times FN)}{\left(TP+FP)(TP+FN)(TN+FP)(TN+FN\right)}$$

True positives, denoted by the symbol (Tp), are accurately predicted positive cases, whereas true negatives (Tn) are accurately predicted negative scenarios. False positives (Fp) are incorrectly predicted positive outcomes, whereas false negatives (Fn) are incorrectly predicted negative outcomes.

Table 4 shows a steady increase in the accuracy, sensitivity, specificity, F1-score, and MCC measures as the number of epochs increases from 10 to 50 and attains a training accuracy of 99.52% and 99.91% after 50 epochs. This was carried out to prove the superiority of the Xception network method: with the smallest number of epochs, the training and validation accuracy are very high. From the table, it can be seen that in comparison with other conventional generic convolutional neural networks, the model does not perform channel-wise convolution, which has the effect of reducing the number of connections and thereby making the model lighter, and hence only a few epochs can bring excellent results in accuracy. As shown on the table, training accuracy is 96.0% with just 10 epochs by the proposed DR-NASNet architecture. In addition, as shown in Table 4, we also compared the original NASNet architecture with the proposed DR-NASNet model. Compared with the original architecture, the proposed DR-NASNet model achieved the highest performance in terms of accuracy. This is due to the fact that we improved the NASNet architecture by integrating dense blocks.

### Computational Cost

According to the complexity of computations, state-of-the-art DL models and the suggested DR-NASNet system were also compared. According to Table 5, the suggested DL architecture required a total processing time of about 184.5 s, where the total processing times were 246.2, 230.1, 217.4, 211.8, 207.5, 195.7, and 193.4 s for the VGG16, AlexNet, InceptionV3, GoogleNet, Xception, MobileNet, and SqueezeNet, respectively. Accordingly, our suggested DR-NASNet technique took less time to identify several severity levels of DR, which is essential in a setting where computational performance is crucial. This demonstrates how effective the suggested idea is in relation to the current paradigm. In addition, we performed curtained experiments, which are explained in the subsequent paragraphs. In addition, the computational speed of the modified NASNet model is increased compared with the original architecture of the NASNet model, as shown in Table 5.

**Experiment** **1.**
*In this experiment, the DR dataset available at Kaggle is used. It consists of five classes. We may infer from Table 5 that the average accuracies for ResNet, GoogleLeNet, and AlexNet were 92.40%, 93.75%, and 94.62%, respectively. The proposed model, however, claimed an accuracy level of roughly 96.61% with preprocessing steps. As a result, it is a reliable tool for identifying and categorizing Diabetic Retinopathy. The data are also divided into training and testing data. The remaining 20% of the data were utilized to evaluate the suggested architecture, while the remaining 80% of the data were used for training. The refined version of VGGNet performed in an incredibly respectable manner. With each new epoch, the rate of training and validation accuracy increased steadily. According to the loss curve, training and validation losses became smaller with each epoch. The augmented data were obtained through the application of several data augmentation techniques. As indicated in Figure 7, the enhanced data collected were divided into training and testing data.*


**Experiment** **2.**
*In this experiment, another dataset from Kaggle called Diabetic Retinopathy 224 × 224 Gaussian Filtered is used. The photos are retina scan images that were Gaussian filtered to find DR. APTOS 2019 Blindness Detection provides access to the original dataset. These photos were downsized to 224 × 224 pixels so that numerous deep learning models that have already been trained can use them with ease. As indicated in Figure 8, the enhanced data collected were divided into training and testing data.*


**Experiment** **3.**
*In this experiment, the images consist of retinal fundus images to detect DR. APTOS 2019 Visual Impairment Detection has an accessible initial dataset. These photos were downsized to 224 × 224 pixels so that several pretrained deep learning models may easily utilize them. As indicated in Figure 9, the enhanced data collected were divided into training and testing data.*


**Experiment** **4.**
*In this study, the Aptos 2019 augmented dataset is considered. The dataset consists of five classes; each class has 2000 images. The dataset is generated using the collected APTOS 2019 dataset along with augmentation methods to increase the number of images for each class. As indicated in Figure 10, the enhanced data collected were divided into training and testing data.*


**Experiment** **5.**
*In this experiment, a novel dataset is proposed (PAK-DR). The dataset is divided into five classes, as shown in Table 6. The suggested building was compared using five modern techniques to determine its strength. This comparison shows the difference in accuracy of the proposed model with different datasets. As can be seen from the Figure, the proposed model achieves the highest accuracy with the PAK-DR dataset. This is because the proposed dataset is well-organized under the supervision of professional ophthalmologists and eye specialists. That means that the proposed dataset has no bugs and is a well-organized dataset as compared with other datasets.*


The proposed model achieves around 99.6% accuracy with the collected dataset, which is the highest accuracy compared with the traditional methods. Several papers in the literature used their own datasets. However, the proposed model achieves the best accuracy. The model is also compared using the resized version of the APTOS competition dataset. The following table shows the relationship between the proposed model and other methods mentioned in the literature.

## 5. Discussion

Diabetes-related Retinopathy (DR) is an eye disease that is the primary cause of blindness. It is a diabetic complication that results from damage to the retinal vasculature. Patients only become aware of this silent disease when they develop vision problems. However, this occurs when retinal alterations have progressed to the stage where vision loss is more likely, and treatment is complex [21]. This disease is irreversible and causes blindness in diabetic patients. However, early DR findings may aid physicians in preventing its progression in diabetic patients. Numerous researchers are motivated to develop automatic recognition systems for DR in order to diagnose the disease early and halt its progression. Retinal vasculature is harmed by DR, and retinal damage arises from microblood vessel destruction brought on by high blood pressure, which causes impaired vision issues. According to the World Health Organization (WHO), there will be 350 million cases of diabetes worldwide during the next 25 years [22]. According to the National Eye Institute Database of the United States, diabetes is a substantial cause of visual impairment in people between the ages of 20 and 74 [23].

From the aforementioned comparison analysis, it can be concluded that the suggested model is a reliable and effective way to identify DR early on. The BenGraham-CLAHE approach has been used in this study’s preprocessing of DR pictures to enhance the image contrast. The NASNet model, which uses a straightforward CNN model as a feature extraction method, has been utilized for dimensionality reduction. Finally, SVM has been employed as the classifier to reduce the difficulty and expense of the training process. The freshly created framework of this suggested model asserts the originality of this investigation. The dataset is not sufficiently big. The model performance may vary with a huge dataset, which is not taken into account in this study. After all, the quality of the images and the preprocessing methods used have a significant impact on model performance. The DR pictures utilized in this investigation are of a rather high caliber. The model’s performance with poor-quality photos is outside the purview of this investigation. In the future, a sizable dataset with a mix of low- and high-quality photos may be used to analyze the model’s performance.

The rate of diabetes is growing in both developed and developing nations. Developed countries have a higher rate of diabetes than undeveloped countries. Several people with DR live in developing countries [24]. This is because of insufficient treatment and improper health care management. Identifying a patient’s disease before treatment is a complicated process.

DL algorithms have shown superior performance for different disease classification tasks. A new methodology is proposed in this paper for DR classification. The methodology depends on a machine learning method called GRAD-CAM, introduced in [25], which is implemented on various available datasets. This technique uses the gradients between the classification score and the final convolutional feature map to pinpoint the areas of an input image that influence the classification score the most. The proposed methodology also depends on a deep learning model, NASNet, applied to DR classification for the first time. The proposed model was enhanced to produce better accuracy by adding dense blocks. Figure 2 shows the improved NASNet model block diagram.

The proposed methodology was used to build a complete automatic DR recognition system. The proposed system is tested using five different datasets. The authors proposed one dataset, and there are four of these datasets online on Kaggle. The model is evaluated and compared with other traditional models mentioned in previous research. The system achieves very high accuracy with the proposed dataset, as it was collected carefully with no bugs or missing data and without noise. Also, in collecting the proposed dataset, a team of professional ophthalmologists and eye specialists organized and labeled the dataset to ensure high accuracy in labeling. For all these reasons, it is clear from the experiments section that the system achieves a maximum accuracy of 99.55%. The comparison section shows the system’s accuracy using different available online datasets. These accuracies are used in the state-of-the-art comparison section to compare the proposed methodology with other techniques introduced in the literature.

It can be seen from the comparisons that the proposed system achieves higher accuracy than the APTOS dataset. Also, the system achieves around 100% accuracy on the proposed dataset. From the previous comparison, it can be evaluated that not only the proposed methodology achieves the highest accuracy on the authors’ proposed dataset (achieving almost 100%), but the model also has the highest accuracy using the APTOS online dataset, i.e., 98.63%. There are two main ideas behind the creation of an automated validation tool based on deep learning that could get rid of lighting problems in retinal fundus images. The first objective was to reduce the amount of human effort required to extract manual features for diagnosis by relying on artificial intelligence and image processing techniques.

The second was the adaptability of deep learning models to a variety of problems, as well as the availability of optimization techniques, such as various regularization techniques, to improve performance. The experimental results on unseen test data in Table 5 show that our main goal was also to lower the number of false negatives and conduct a performance evaluation of comparable works on the multiclass fundus dataset class distribution of the Kaggle multiclass dataset [5].

Only 2.2% of false negatives demonstrate the accuracy of our model. Our method is also economically feasible to implement because it does not require expensive hardware or gadgets with a powerful graphical processing unit (GPU). According to [7], DR detection sensitivity values larger than 60 percent are cost-effective. Due to the fact that our model was trained with a dataset containing numerous variations, it also demonstrates the high adaptability and robustness of our model to perform precisely with nonideal illumination fundus images. In Table 5 and Table 6, the proposed model’s performance on binary classification is compared with previous work on DR detection using similar multsource datasets. Multiple binary and multiclass datasets were utilized to validate the results of our proposed model. It can be observed that [8] achieved an accuracy of 97.30%, but their model is less sensitive than our proposed layered model. Therefore, our model is superior at accurately detecting true positives.

In another study, two deep learning models were employed for Diabetic Retinopathy (DR) detection and classification: a hybrid network consisting of VGG16, XgBoost Classifier, and DenseNet 121, and a standalone DenseNet 121 model [26]. The evaluation was conducted on retinal images from the APTOS 2019 Blindness Detection Kaggle Dataset, with suitable balancing techniques applied to address the class imbalance. The results show that the hybrid network achieved an accuracy of 79.50%, while the DenseNet 121 model performed significantly better with an accuracy of 97.30%. Also, comparing the DenseNet 121 model with other methods using the same dataset showed that it was better, showing how well it works at finding and classifying DR early on. However, the DL [27] models show that the accuracy is greatly improved compared with machine learning techniques. Therefore, this paper utilizes a modified deep learning model based on the NASNet architecture.

The performance of our model for multiclass classification on the Kaggle dataset is compared in Table 6. According to Table 5, the dataset contains five stages of DR: healthy, mild, moderate, severe, and advanced. With a final test accuracy of 87.45%, the proposed model outperformed all other models by achieving the highest sensitivity and specificity values. Due to the imbalanced data depicted in the provided dataset, this accuracy score is inferior to the binary classification results. Table 6 demonstrates the accuracy and precision of the proposed model on multiple binary and multiclass datasets. Taking into account all of the metrics from Figure 8, Figure 9 and Figure 10, we can say that the proposed model outperforms state-of-the-art models and works well for both binary and multiclass classification of DR. Moreover, Figure 11 indicates that the proposed model accurately detected the severity level of DR from retinograph images. Also, Figure 12 shows GRAD-CAM patterns identified by a proposed system when diagnosed by retinograph images.

## 6. Conclusions and Future Works

Millions of people throughout the world suffer from DR. The early detection of the disease facilitates the prevention of DR. In this research, we propose a fully automated detection and classification system for DR. The model is built using a combination of novel preprocessing stages and deep learning models to achieve the best accuracy. The technique uses Ben Graham’s concept and CLAHE image preprocessing techniques to lessen noise, emphasize lesions, and improve DR classification performance. The novel image processing technique proposed in this paper is the Grad-CAM technique, which was used to highlight the locations on DR-effected images that are particularly essential in identifying their presence. The modified deep learning model of NASNet is also proposed in this research work. The model is evaluated using four different available datasets and the authors’ collected dataset (PAK-DR), which includes different datasets obtained from different sources. The state-of-the-art comparison shows the superior performance of the proposed methodology with respect to previous models discussed in the literature. A comparison of their respective strengths and limitations demonstrates that the proposed methodology outperforms the current established models. To demonstrate the efficacy of the proposed method, it must be evaluated on a large and complex dataset, ideally including a substantial number of prospective DR instances. New datasets may be analyzed using DenseNet, VGG, or ResNet, in addition to other augmentation techniques.

This is a prescreening and automatic method to recognize the severity and level of diabetes. It is not a decision support system. However, in the future, we can include some knowledge-based information to develop and extend this DR-NASNet model to provide us with information about the issue of uncontrolled blood glucose levels in diabetic patients. Also, sensitivity to image quality will be addressed in future research to ensure its widespread applicability and reliability in diverse clinical settings.

## Figures and Tables

**Figure 1 diagnostics-13-02645-f001:**
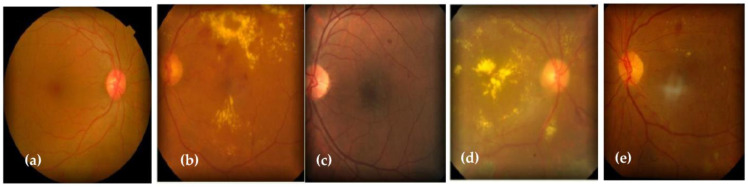
Five stages of Diabetic Retinopathy: (**a**) normal, (**b**) moderate, (**c**) mild, (**d**) proliferative DR, and (**e**) severe DR.

**Figure 2 diagnostics-13-02645-f002:**
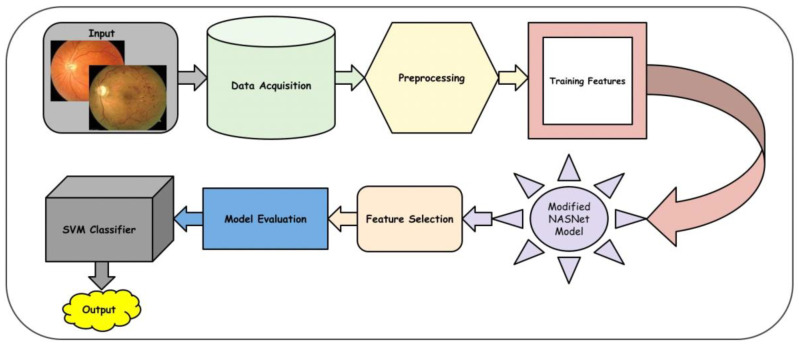
A Systematic flow diagram of a proposed DR-NASNet system.

**Figure 3 diagnostics-13-02645-f003:**
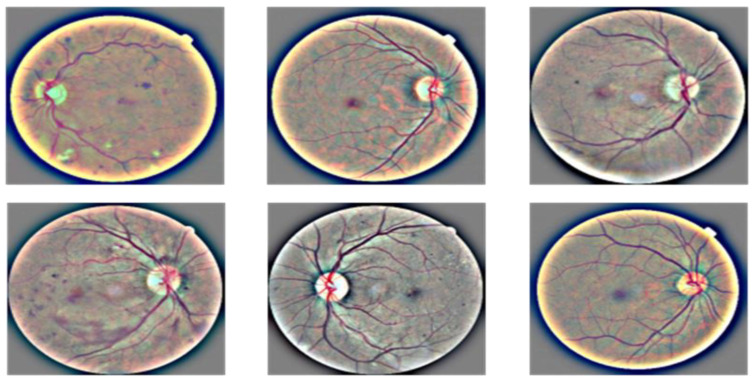
A visual result of image preprocessing to enhance the contrast while adjusting noise pixels.

**Figure 4 diagnostics-13-02645-f004:**
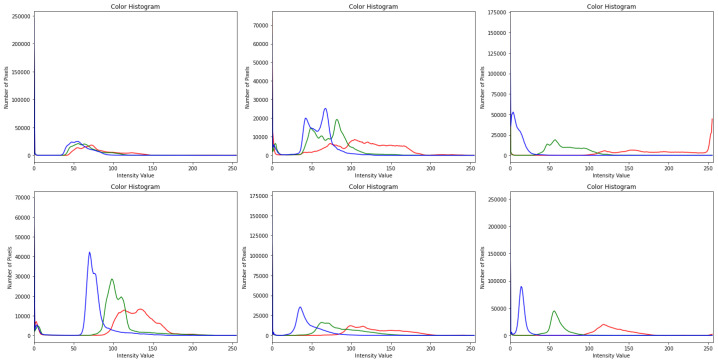
Pixels distribution by using color histogram after contrast adjustment by using CLAHE and Ben Graham in dermoscopy images with respect to Figure 3, where blue color lines indicate the results obtained by CLAHE+ Ben Graham, green lines show the Ben Graham, and red lines show the CLAHE techniques.

**Figure 5 diagnostics-13-02645-f005:**
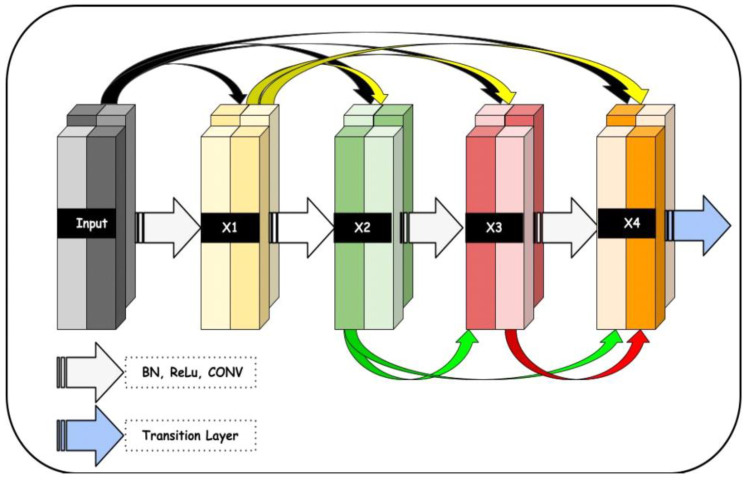
A proposed dense block architecture utilized in the development of the proposed system, where the input size of the retinograph image is (224 × 224 × 3) pixels.

**Figure 6 diagnostics-13-02645-f006:**
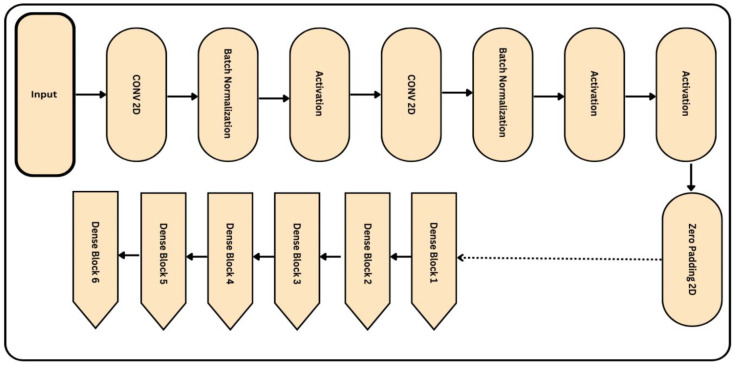
Proposed architecture of the improved NASNet model, where the input size of the retinograph image is (224 × 224 × 3) pixels.

**Figure 7 diagnostics-13-02645-f007:**
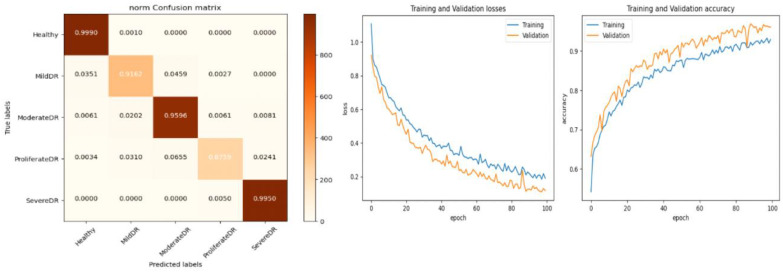
Experimental results produced by experiment 1 and training and validation loss function with result to accuracy.

**Figure 8 diagnostics-13-02645-f008:**
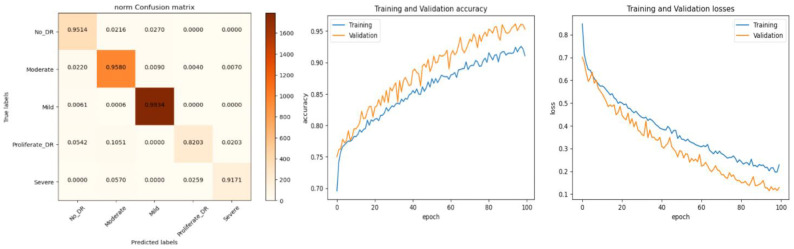
Experimental results produced by experiment 1 and training and validation loss function with result to accuracy.

**Figure 9 diagnostics-13-02645-f009:**
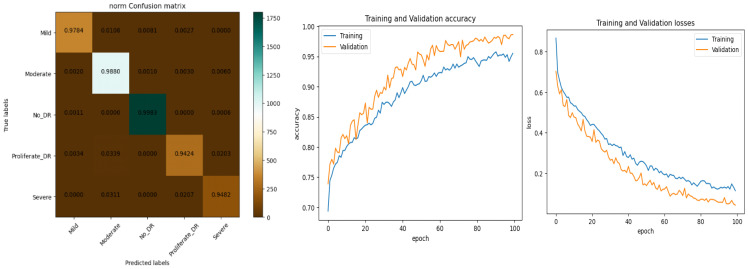
Experimental results produced by experiment 1 and training and validation loss function with result to accuracy.

**Figure 10 diagnostics-13-02645-f010:**
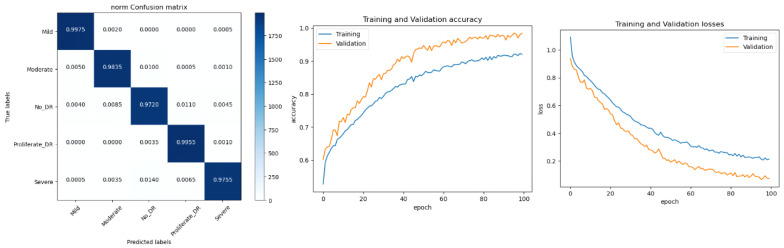
Experimental results produced by experiment 1 and training and validation loss function with result to accuracy.

**Figure 11 diagnostics-13-02645-f011:**
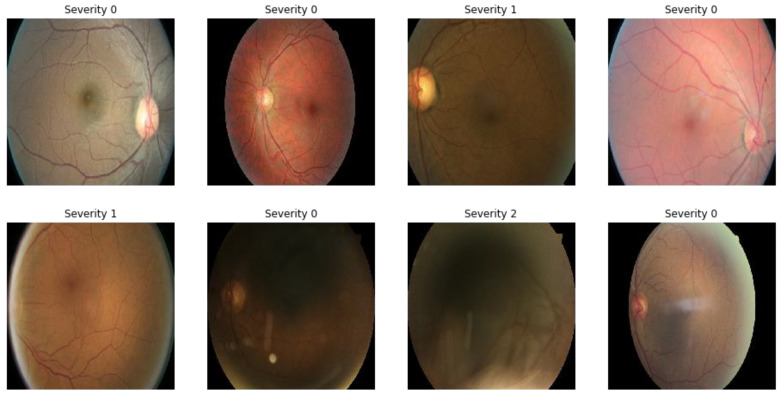
A visual example of recognized DR severity of retinographies.

**Figure 12 diagnostics-13-02645-f012:**
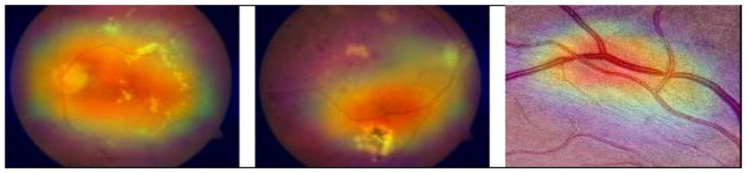
Patterns identified severity levels of Diabetic Retinopathy when diagnosis by retinograph images.

**Table 1 diagnostics-13-02645-t001:** Accuracies and recalls of various methods for DR classification.

Authors	Year	Methodology	Accuracy	Recall
Al-Antary [13]	2021	Multiscale Attention Network (MSA-Net)	APTOS 84.6% Eyepacs 87.5%	APTOS 91% Eyepacs 90.6%
Majumedar [14]	2021	Squeeze Excitation Densely Connected Deep CNN	Not reported	APTOS 96% Eyepacs 93%
Macsik [12]	2022	Local Binary Convolutional Neural Network (LBCNN)	APTOS 97.41%	APTOS 94.63%
Saranya [16]	2022	Support Vector Machine (SVM)	APTOS 94.5% IDRiD 93.3%	APTOS 75.6% IDRiD 78.6%
Kobat [15]	2022	DenseNet201	APTOS 93.85% ND 94.06%	APTOS 80.6% ND 94.45%
M.M. FARAG [17]	2022	DenseNet169+CBAM	APTOS 97%	-
Hayati [2]	2023	EfficientNet	APTOS 97%	-
Pradeep Kumar Jena [18]	2023	U-net for Segementation CNN with SVM for Classification	APTOS 96.9% MESSIDOR 98.3%	-
S. Zulaikha Beevi [19]	2023	Squeezenet, the Picture is Classed into the Normal or Abnormal DCNN for Severity Level.	91.1%	-

**Table 2 diagnostics-13-02645-t002:** Parameters for data augmentation used in this paper.

Parameters	Values
Image cropping	Images were arbitrarily reduced in size to between 60 and 75 percent of their original size.
Flipping:	The X and Y axes of the images were both flipped.
Translation:	Images have a 0-to-30-pixel shift.
Shearing	Images were sheared at random angles between 0 and 180.
Rotation Angle	Images were rotated in a random direction between 0 and 360 degrees.
GST Augmentation	A two-dimensional picture was subjected to the GST-based augmentation
Dataset Enhancement	Used preprocessing method to enhance contrast by using BenGraham-CLAHE techniques.

**Table 3 diagnostics-13-02645-t003:** The distribution of augmented data into training and testing data.

Grade	Severity	Original Images	Augmented Images	Training	Validation
0	Normal	24,000	24,000	20,000	4000
1	Mild	2200	24,000	18,000	6000
2	Moderate	5000	24,000	21,000	3000
3	Severe	900	24,000	20,000	4000
4	PDR	700	24,000	20,000	4000
	Total	32,800	120,000	99,000	21,000

**Table 4 diagnostics-13-02645-t004:** Classification results in terms of different measures by standard DL algorithms compared with the proposed system.

Sr. No	Architecture	Sensitivity	Specificity	Accuracy	Precision	F1-Score	MCC
1	ResNet	0.734	0.843	0.840	0.807	0.818	0.824
2	CNN+LSTM	0.734	0.843	0.840	0.807	0.818	0.824
3	GoogLeNet	0.755	0.889	0.870	0.880	0.839	0.835
4	AlexNet	0.840	0.838	0.860	0.837	0.845	0.849
5	VGGNet	0.880	0.884	0.845	0.885	0.867	0.871
6	InceptionV3	0.840	0.838	0.860	0.837	0.845	0.849
7	NASNet	0.860	0.858	0.880	0.847	0.855	0.869
**8**	**DR-NASNet**	**0.960**	**0.938**	**0.960**	**0.937**	**0.945**	**0.949**

**Table 5 diagnostics-13-02645-t005:** Average processing time on a PSLs dataset by various DL algorithms.

Method	Preprocessing	Feature Extraction	Training	Prediction	Overall
VGG16	20.5 s	14.4 s	200.5 s	10.8 s	246.2 s
AlexNet	18.6 s	12.2 s	190.5 s	8.8 s	230.1 s
InceptionV3	16.3 s	14.8 s	178.5 s	7.8 s	217.4 s
GoogleNet	17.2 s	17.3 s	170.5 s	6.8 s	211.8 s
Xception	18.1 s	15.1 s	165.5 s	8.8 s	207.5 s
MobileNet	14.1 s	13.3 s	160.5 s	7.8 s	195.7 s
SqueezeNet	10.8 s	8.3 s	168.5 s	5.8 s	193.4 s
NASNet	8.8 s	7.3 s	190.5 s	3.8 s	194.4 s
**Proposed**	**1.8 s**	**1.9 s**	**165.5 s**	**1.5 s**	**184.5 s**

**Table 6 diagnostics-13-02645-t006:** Performance comparison of the proposed framework with baseline methods.

Reference #	Author	Year	Model	Target	Accuracy
[13]	Al-Antary	2021	Multiscale Attention Network (MSA-Net)	DR	84.6%
[14]	Majumedar	2021	Local Binary Convolutional Neural Network (LBCNN)	DR	97%
[12]	Macsik	2022	DenseNet169+CBAM	DR	97%
[16]	Saranya	2022	EfficientNet	DR	97%
[15]	Kobat	2022	U-net for segmentation CNN with SVM for classification	DR	96.9%
[17]	M.M. FARAG	2022	CNN	DR	95.03
[2]	Hayati	2023	CNN	DR	82.2
[18]	Pradeep Kumar Jena	2023	CNN	DR	87.06
[19]	S. Zulaikha Beevi	2023	CNN	DR	86.10
[13]	Al-Antary	2021	DCNN	DR	86.04
	Proposed Method		DR-NASNet	DR	96.05

## Data Availability

Dataset will be available upon request.

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
