# Peer review of "DR-NASNet: Automated System to Detect and Classify Diabetic Retinopathy Severity Using Improved Pretrained NASNet Model"

_diagnostics, 2023, doi:10.3390/diagnostics13162645_

Round 1

Reviewer 1 Report

The primary goal is to create a reliable method for automatically detecting the severity of DR. 

This paper proposes a new automated system (DR-NASNet) to detect and classify DR severity using an improved pretrained NASNet Model. To develop the DR NASNet system, it utilized a preprocessing technique that takes advantage of Ben Graham and CLAHE to lessen noise, emphasize lesions, and ultimately improve DR classification performance.  The DR-NASNet system was tested using six different experiments. The system achieves 96.05% accuracy with the challenging DR dataset. But many works are already available in this domain, so authors should compare them with some existing competitive work as well as methods.

In Table 1 better to cite the reference number with the author's name. In the same table of recall, columns are blank for the last 5 references. All are not used recall?

Many works of literature are available on classification, which may add to the discussion. Some of the recent articles you may take a look at. 

1. . Mohanty, Cheena, et al. "Using Deep Learning Architectures for Detection and Classification of Diabetic Retinopathy." Sensors 23.12 (2023): 5726.

2. Dash, Sujata, et al., eds. Deep learning techniques for biomedical and health informatics. Cham, Switzerland: Springer International Publishing, 2020.

Author Response

Comment - (1) This paper proposes a new automated system (DR-NASNet) to detect and classify DR severity using an improved pretrained NASNet Model. To develop the DR NASNet system, it utilized a preprocessing technique that takes advantage of Ben Graham and CLAHE to lessen noise, emphasize lesions, and ultimately improve DR classification performance.  The DR-NASNet system was tested using six different experiments. The system achieves 96.05% accuracy with the challenging DR dataset. But many works are already available in this domain, so authors should compare them with some existing competitive work as well as methods.

Response 1: Our proposed method was compared against state-of-art models provided in literature. It can be seen in table 5, which shows a comparison between our proposed model and other models form literature that shows clearly the superior performance of the proposed method. Also, in experiment 5 we run the proposed method on a privately collected dataset from Pakistani hospitals. The proposed methodology achieves 99.6% accuracy.

Thank you for the comment.

Comment - (2) In Table 1 better to cite the reference number with the author's name. In the same table of recall, columns are blank for the last 5 references. All are not used recall?.

Response 2: We thank the reviewer for his valuable comment. Reference numbers are added near authors names as requested by reviewer. As for recall values if the values are not reported in referenced text, we leave it as blank.

Thank you for clearing up this problem in our paper.

Comment - (3) Many works of literature are available on classification, which may add to the discussion. Some of the recent articles you may take a look at.

  1. . Mohanty, Cheena, et al. "Using Deep Learning Architectures for Detection and Classification of Diabetic Retinopathy." Sensors 23.12 (2023): 5726.
  2. Dash, Sujata, et al., eds. Deep learning techniques for biomedical and health informatics. Cham, Switzerland: Springer International Publishing, 2020.

Response 3: Yes, you are right.

We thank the reviewer for this. We added these references to the discussion section.

We have added the following paragraph on page#20 and start at line# 940 as:

“In another study, two deep learning models were employed for diabetic retinopathy (DR) detection and classification. A hybrid network consisting of VGG16, XgBoost Classifier, and DenseNet 121, and a standalone DenseNet 121 model [26]. The evaluation was conducted on retinal images from the APTOS 2019 Blindness Detection Kaggle Dataset, with suitable balancing techniques applied to address the class imbalance. The results showed that the hybrid network achieved an accuracy of 79.50%, while the DenseNet 121 model performed significantly better with an accuracy of 97.30%. Furthermore, the comparison with existing methods using the same dataset revealed the superiority of the DenseNet 121 model, highlighting its effectiveness in early detection and classification of DR. However, the DL [27] models show that the accuracy is greatly improved compared to machine-learning techniques. Therefore, this paper utilized a modified deep learning model based on NASNet architecture.

Thank you for this valuable comment to increase the readability of the manuscript.

Reviewer 2 Report

This paper proposed a new automated system (DR-NASNet) to detect and classify DR severity using an improved pretrained NASNet Model.

1. What is NASNet? If the paper said improved pretrained NASNet model, it means there is previous work for NASNet. Please add citations to other mentioned methods too to give respect to previous studies.

2. The reviewer encourages the author to compare the original NASNet vs the improved NASNet with the same data training, validating, and testing. Then report the result. This suggestion proves that the statement of the paper title where using improved pretrained NASNet is correct.

3. The reviewer wants to comment on the statement "This paper proposes a new automated system (DR-NASNet) to detect and classify DR severity using an improved pretrained NASNet Model.". What kind of result from the paper can answer the question "to detect"? The reviewer agrees that the paper has answered "to classify".

4. In Figures 5 and 6, please give the size of the input and output for every block. For example, the input is the image (256, 256, 3), and so on. It is a good way to understand well your architecture.

The English are fine.

Author Response

Comment - This paper proposed a new automated system (DR-NASNet) to detect and classify DR severity using an improved pretrained NASNet Model.

  1. What is NASNet? If the paper said improved pretrained NASNet model, it means there is previous work for NASNet. Please add citations to other mentioned methods too to give respect to previous studies.”

Response 1: As suggested by reviewer #2, we have updated all those sentences in the revised manuscript.

NASNet is a type of convolutional neural network discovered through neural architecture search. The paper is first attempt to apply NasNet networks to the problem of DR classification. However, the authors suggests adding dense blocks to the architecture to increase its accuracy over ordinary NasNet networks.

To show this change, we have revised our major contribution section as:

  • A modified NASNet methodology is proposed for classification of severity level of diabetic retinopathy.
  • The suggested methodology is trained and tested on a novel dataset, which is itself a big challenge.
  • The use of several preprocessing methods to enhance the usefulness and visibility of retinal images using histogram equalization and Ben Graham's processing technique.

Thank you for clearing up this problem in our paper.

Comment - (2) The reviewer encourages the author to compare the original NASNet vs the improved NASNet with the same data training, validating, and testing. Then report the result. This suggestion proves that the statement of the paper title where using improved pretrained NASNet is correct.

Response 2: Yes, you are right. In the revised version of the paper, we have compared the original NASNet with the modified one. Thank you to point out this viewpoint.

Following changes can be observed as:

Table 4. Classification results in terms of different measures by standard DL algorithms compared to proposed system

Sr. No

Architecture

Sensitivity

Specificity

Accuracy

Precision

F1-Score

MCC

1

ResNet

0.734

0.843

0.840

0.807

0.818

0.824

2

CNN+LSTM

0.734

0.843

0.840

0.807

0.818

0.824

3

GoogLeNet

0.755

0.889

0.870

0.880

0.839

0.835

4

AlexNet

0.840

0.838

0.860

0.837

0.845

0.849

5

VGGNet

0.880

0.884

0.845

0.885

0.867

0.871

6

InceptionV3

0.840

0.838

0.860

0.837

0.845

0.849

7

NASNet

0.860

0.858

0.880

0.847

0.855

0.869

8

DR-NASNet

0.960

0.938

0.960

0.937

0.945

0.949

.

4.4 Computational Cost

According to complexity of computations, state-of-the-art DL models and the suggested DR-NASNet system were also compared. According to Table 5, the suggested DL architecture required a total processing time of about 184.5 seconds. Where the total processing times were 246.2, 230.1, 217.4, 211.8, 207.5, 195.7, and 193.4 seconds for the VGG16, AlexNet, InceptionV3, GoogleNet, Xception, MobileNet, and SqueezeNet, respectively. Accordingly, our suggested DR-NASNet technique took less time to identify several severity-level of DR, which is essential in a setting where computational performance is crucial. This demonstrates how effective the suggested idea is in relation to the current paradigm. In addition, we have performed curtained experiments, which are explained in the subsequent paragraphs.

Table 5. Average processing time on a PSLs dataset by various DL algorithms.

Method

Preprocessing

Feature

Extraction

Training

Prediction

Overall

VGG16

20.5s

14.4s

200.5s

10.8s

246.2s

AlexNet

18.6s

12.2s

190.5s

8.8s

230.1s

InceptionV3

16.3s

14.8s

178.5s

7.8s

217.4s

GoogleNet

17.2s

17.3s

170.5s

6.8s

211.8s

Xception

18.1s

15.1s

165.5s

8.8s

207.5s

MobileNet

14.1s

13.3s

160.5s

7.8s

195.7s

SqueezeNet

10.8s

8.3s

168.5s

5.8s

193.4s

NASNet

8.8s

7.3s

190.5s

3.8s

194.4s

Proposed

1.8s

1.9s

165.5s

1.5s

184.5s

Comment - (3) The reviewer wants to comment on the statement "This paper proposes a new automated system (DR-NASNet) to detect and classify DR severity using an improved pretrained NASNet Model.". What kind of result from the paper can answer the question "to detect"? The reviewer agrees that the paper has answered "to classify".?

Response 3: Yes, you are right. The authors intention from the word detect is that if the model classifies the image into images with disease and images without disease then it means the model detected the disease. However, we removed the detect word from manuscript.

Thank you for these valuable comments.

Comment - (4) In Figures 5 and 6, please give the size of the input and output for every block. For example, the input is the image (256, 256, 3), and so on. It is a good way to understand well your architecture.

Response 4: As suggested by reviewer Reviewer# 2, we have updated those citated figures captions as:

The input is the image (256, 256, 3), and so on.

Figure 5. A proposed dense block architecture utilized in the development of proposed system, where input size of retinograph image is (224×224×3) pixels.

Figure 6. Proposed Architecture of Improved NASNet Model, where input size of retinograph image is (224×224×3) pixels.

Thank you for this valuable comment.

Reviewer 3 Report

I have only intentions for the readability and reproducibility of the study. The following questions can be addressed:

What are the main challenges associated with detecting and classifying the severity of Diabetic Retinopathy (DR)?

Is it a binary or multi-class classification? Please clarify and make changes as necessary.

How does the proposed DR-NASNet system address the issue of uncontrolled blood glucose levels in diabetic patients?

What preprocessing techniques are utilized in the development of the DR-NASNet system, and how do they improve DR classification performance?

How does the integration of dense blocks in the DR-NASNet system enhance the effectiveness of classification results for different severity levels of DR?

What datasets are used to test the performance of the DR-NASNet system, and how do they contribute to the learning of effective features from DR images?

What evaluation measures are used to assess the performance of the DR-NASNet system, and how does it compare to existing methods?

How does the addition of the classifier layer of a linear SVM with a linear activation function contribute to the final classification of DR severity levels?

Can the high accuracy achieved by the DR-NASNet system be generalized to other challenging DR datasets?

What are the practical implications of the DR-NASNet system for ophthalmologists and their ability to classify early-stage levels of DR?

Are there any limitations or potential drawbacks associated with the proposed DR-NASNet system, and how could they be addressed in future research?

Please proofread once again. Many tables are wrongly mentioned, and events do not exist (table 8).

Could the source code of the model be released? It is hard to justify its correctness without checking the source code.

I strongly suggest language editing since it is difficult to understand what the authors want to express. 

Author Response

Comment -1 What are the main challenges associated with detecting and classifying the severity of Diabetic Retinopathy (DR)?”

Response 1: As suggested by reviewer #3, we have added and updated all those sentences in the revised manuscript as:

Detecting and classifying the severity of Diabetic Retinopathy (DR) is a complex task that involves several challenges, some of which include:

  1. Image Quality
  2. Large Dataset Annotation
  3. Class Imbalance
  4. Inter-observer Variability
  5. Feature Extraction
  6. Privacy and Data Security
  7. Clinical Validation

This information has been added to the section 1.1 research motivations and challenges section on page#3.

Thank you for clearing up this problem in our paper.

Comment -2 Is it a binary or multi-class classification? Please clarify and make changes as necessary.

Response 2: It is a multi-class classification problem to recognize five stages of Diabetic retinopathy.

The final stage is to classify the image as No DR, Mild, Moderate, Proliferate, or Severe, which requires the addition of the classifier layer of linear SVM with a linear activation function.

Thank you for the comment.

Comment -3 How does the proposed DR-NASNet system address the issue of uncontrolled blood glucose levels in diabetic patients?

Response 3: Yes, you are right.

It is a pre-screening and automatic method to recognize severity-level of diabetes. It is not a decision-support system. However, in the future works, we can include some knowledge-based information to develop and extend this DR-NASNet model to provide us information about the issue of uncontrolled blood glucose levels in diabetic patients.

We have added this information in the updated version under the heading of Conclusion and future works on page# 20.

Thank you for the comment.

Comment -4 What preprocessing techniques are utilized in the development of the DR-NASNet system, and how do they improve DR classification performance?

Response 4: BenGraham and CLAHE are preprocessing techniques used to enhance retinal images in Diabetic Retinopathy (DR) classification. BenGraham normalizes color channels, increasing contrast, vessel visibility, and reducing noise. CLAHE enhances local contrast, corrects illumination, and preserves texture. Both techniques improve feature extraction, leading to better DR severity classification.

Thank you for the comment.

Comment -5 How does the integration of dense blocks in the DR-NASNet system enhance the effectiveness of classification results for different severity levels of DR?

Response 5: Yes, this information is added to the research highlights to show the advantage of proposed DR-NASNet model.

We have added the following information as:

Integrating dense blocks in the NASNet system enhances DR classification by enabling efficient feature reuse, mitigating vanishing gradients, and promoting multi-scale feature fusion. The model learns rich representations, improving classification results for different DR severity levels with fewer parameters and better pattern detection.

“.

Thank you for the comment.

Comment -6 What datasets are used to test the performance of the DR-NASNet system, and how do they contribute to the learning of effective features from DR images?

Response 6: Dear reviewer#3, we have already added a separate section on page#7 and start at line# 330 as:

3.1 Data Acquisition

This procedure entails compiling data from numerous sources, putting it in an appropriate format, and archiving it for future use [20]. Dataset acquisition must fol-low strict protocols to ensure that the quality of the images is satisfactory for analysis. Factors like image resolution, illumination, contrast, and colour balance must be con-sidered to ensure consistency across the dataset. This is the first step in the proposed model, which is very critical for the performance of the CAD-DR system. After acquisi-tion, the images are resized to the appropriate sizes, which are suitable for the deep learning model.

Thank you for the comment.

Comment -7 What evaluation measures are used to assess the performance of the DR-NASNet system, and how does it compare to existing methods?

Response 7: The performance of the DR-NASNet system for DR classification is evaluated using measures like accuracy, sensitivity, specificity, precision, F1 score and AUC-ROC. It is compared against existing methods, demonstrating its advantages in feature extraction, representation learning, and parameter efficiency, leading to potentially higher accuracy and improved DR severity classification.

Comment -8 How does the addition of the classifier layer of a linear SVM with a linear activation function contribute to the final classification of DR severity levels?

Response 8: Yes, this information is added to the research highlights to show the advantage of proposed DR-NASNet model.

We have added the following information as:

Adding a linear SVM classifier with a linear activation function contributes to the final classification of DR severity levels by creating decision boundaries, providing confidence scores, handling multi-class tasks, and leveraging robustness and interpretability. It complements deep features from models like DR-NASNet, leading to accurate predictions for different Diabetic Retinopathy stages.

Thanks for your valuable comments.

Comment -9 Can the high accuracy achieved by the DR-NASNet system be generalized to other challenging DR datasets?

Response 9 Generalization depends on various factors and requires rigorous evaluation, transfer learning, and domain adaptation techniques to improve its suitability for different datasets. No complete guarantee of generalization is possible in the context of machine learning models.

Thanks for your valuable comments.

Comment -10 What are the practical implications of the DR-NASNet system for ophthalmologists and their ability to classify early-stage levels of DR?

Response 10: The DR-NASNet system has practical implications for ophthalmologists, as it allows for early detection, efficient triage, reduced workload, and standardized diagnosis of early-stage Diabetic Retinopathy. It enables telemedicine and remote care, supports research, aids in education and training, and enhances overall patient outcomes.

Thanks for your valuable comments.

Comment -11 Are there any limitations or potential drawbacks associated with the proposed DR-NASNet system, and how could they be addressed in future research?

Response 11: Sensitivity to image quality, must be addressed in future research to ensure its widespread applicability and reliability in diverse clinical settings.

This information has been updated in the conclusion section on page#20 as:

“It is a pre-screening and automatic method to recognize severity-level of diabetes. It is not a decision-support system. However, in the future works, we can include some knowledge-based information to develop and extend this DR-NASNet model to pro-vide us information about the issue of uncontrolled blood glucose levels in diabetic pa-tients. Also, sensitivity to image quality, will be addressed in future research to ensure its widespread applicability and reliability in diverse clinical settings.

Thanks for your valuable comments.

Comment -12 Please proofread once again. Many tables are wrongly mentioned, and events do not exist (table 8).?

Response 12: Yes, you are right. We have proof read whole papers and improves its readability. Those changes can be seen by using word tracking software.

Thanks for your valuable comments.

Comment -13 Could the source code of the model be released? It is hard to justify its correctness without checking the source code?

Response 13: Yes, we can provide the code. However, if some authors requested through proper channel then we can provide them.

Thanks for your valuable comments.

Comment -14 I strongly suggest language editing since it is difficult to understand what the authors want to express.?

Response 14: Yes, you are right. We have proof read whole papers and improves its readability. Those changes can be seen by using word tracking software.

Thanks for your valuable comments.